# Biochemical Defence of Plants against Parasitic Nematodes

**DOI:** 10.3390/plants13192813

**Published:** 2024-10-08

**Authors:** Birhanu Kahsay Meresa, Jasper Matthys, Tina Kyndt

**Affiliations:** 1Biotechnology Department, Mekelle University, Mekelle P.O. Box 231, Ethiopia; birhanu.kahsay@mu.edu.et; 2Biotechnology Department, Ghent University, B-9000 Ghent, Belgium; jasper.matthys@ugent.be

**Keywords:** nematodes, biochemical defence, NAMP receptor, cell wall reinforcement, antioxidants, phytochemicals, transcription factor, epigenetics

## Abstract

Plant parasitic nematodes (PPNs), such as *Meloidogyne* spp., *Heterodera* spp. and *Pratylenchus* spp., are obligate parasites on a wide range of crops, causing significant agricultural production losses worldwide. These PPNs mainly feed on and within roots, impairing both the below-ground and the above-ground parts, resulting in reduced plant performance. Plants have developed a multi-component defence mechanism against diverse pathogens, including PPNs. Several natural molecules, ranging from cell wall components to secondary metabolites, have been found to protect plants from PPN attack by conferring nematode-specific resistance. Recent advances in *omics* analytical tools have encouraged researchers to shed light on nematode detection and the biochemical defence mechanisms of plants during nematode infection. Here, we discuss the recent progress on revealing the nematode-associated molecular patterns (NAMPs) and their receptors in plants. The biochemical defence responses of plants, comprising cell wall reinforcement; reactive oxygen species burst; receptor-like cytoplasmic kinases; mitogen-activated protein kinases; antioxidant activities; phytohormone biosynthesis and signalling; transcription factor activation; and the production of anti-PPN phytochemicals are also described. Finally, we also examine the role of epigenetics in regulating the transcriptional response to nematode attack. Understanding the plant defence mechanism against PPN attack is of paramount importance in developing new, effective and sustainable control strategies.

## 1. Introduction

Plants are constantly exposed to biotic stresses such as parasitic nematodes, which impair their productivity and lead to significant agricultural production losses worldwide. To date, more than 4100 plant parasitic nematode (PPN) species have been described as important restraints to agricultural productivity [1], causing estimated yield losses ranging from 5% up to 20% and valuing USD 175–200 billion worldwide [2]. PPNs are categorised as ectoparasitic and endoparastic nematodes based on their feeding style. Endoparasitic nematodes spend at least part of their life inside the host, most often in the root tissue to feed, whereas ectoparasites feed from the outside. PPN attack and feeding cause tissue damage and necrosis as they take away nutrients and sugars, leaving the plant weaker than before. In turn, plants use their constitutive or/and induced defence mechanisms to withstand PPN parasitism. Pre-formed structural barriers and phytoanticipins are examples of the constitutive defence mechanisms that can make a plant ‘non-host’ by preventing the nematodes from invading [3,4]. Inducible defence mechanisms, such as phytoalexins, are activated following PPN penetration [4].

Revealing plant defence mechanisms against PPNs is of paramount importance in developing new, effective and sustainable control strategies for PPN management. Infective nematodes rely on plant-released attractant metabolites to locate suitable hosts under natural conditions [5]. Host plants could detect approaching PPNs before they make physical contact by sensing PPN-originated compounds and initiating defence responses, similar to the detection of pathogen-associated molecular patterns or PAMPs [6]. Ascarosides (pheromone derivatives of dideoxysugar ascarylose) released by different PPNs have been described as nematode-associated molecular patterns (NAMPs) that can be detected by the host plant using their surface-localised pattern recognition receptors (PPRs) [6]. Upon contact, PPN infection causes damage to the plant tissues, leading to the release of damage-associated molecular patterns (DAMPs) that can trigger wounding-related plant defence responses [6]. Following the detection of NAMPs or DAMPs, plants respond by inducing various biochemical changes associated with stress signalling that cause the activation of pattern-triggered immunity (PTI) as the first line of inducible defence against an invading nematode. On the other hand, PPNs could take dominance over PTI by producing effectors that counteract PTI responses [7].

Some varieties carry nematode resistance genes, such as the *Mi-1.2* gene that makes specific tomato cultivars resistant to root-knot nematodes, such as *Meloidogyne incognita* [8]. Resistance proteins recognise nematode effectors leading to induction of effector-triggered immunity (ETI), which is often systemic and important to acquire a strong defence response [9]. Induced cellular defence activities like the reactive oxygen species (ROS) burst, cell wall reinforcement, kinase-dependent signalling, phytohormone production, transcription factor (TF) activation and pathogenesis-related (PR) protein synthesis are involved in both PTI and ETI [10]. Interconnections of these activities are crucial for enhancing immune responses not only at locally infected but also at distal sites, thereby restricting systemic pathogen/pest spread [10].

As a result of these early signalling events, metabolites with anti-nematode activities are being produced [4]. The defensive metabolites with described activity against PPN include enzymatic antioxidants, phenolic compounds, organosulphur compounds, terpenoids, alkaloids, saponins, benzoxazinoids and glucosinolates [4]. Some of these metabolites function as phytoanticipins, and some of them are phytoalexins [4]. Over 2 billion secondary metabolites have been discovered in the plant kingdom [11], and thus, a detailed investigation of their role during plant–PPN interaction could pave the way in the search for novel anti-PPN compounds.

With the advancement of *omics* techniques, plant nematology experts have devoted themselves to deciphering plant defence mechanisms against PPN attacks. In this review, we summarise the recent findings on how dicot and monocot plants recognise PPNs and the biochemical defence mechanisms of these plants in response to invasion and feeding by PPNs such as *Meloidogyne* spp., *Heterodera* spp. and *Pratylenchus* spp. After describing the perception of nematode presence in the plant, we highlight the intracellular signalling events and the downstream effects on the epigenome, transcriptome and finally the metabolome.

## 2. Nematode Perception by the Plant

### 2.1. Perception at the Plasma Membrane

Plants sense pathogens by detecting PAMPs through their PRRs, resulting in the initiation of PTI in the host. For example, chitin and β-glucan from fungi and peptidoglycan, flagellin, elongation factor Tu and lipopolysaccharide from bacteria are well-conserved PAMPs [10]. However, information about PPN-originated PAMPs and their potential receptors in host plants remains scarce. Ascarosides are pheromones secreted by PPN species that play important roles in their reproduction, growth and host infection. Ascaroside #18 (ascr#18) is produced by most PPN species and is currently the only NAMP that is known to activate PTI in a broad spectrum of host phytopathogen systems [12,13]. Exogenously supplied ascr#18 provides protection to Arabidopsis (*Arabidopsis thaliana*) against *Heterodera schachtii* and *M. incognita* and other pathogens such as *Pseudomonas syringae* through activation of mitogen-activated protein kinase (MAPK), jasmonic acid (JA) and salicylic acid (SA) pathways [14]. Ascarosides are found in NemaWater, a solution created by incubating second-stage juvenile (J2) nematodes in water for 24 h and then removing them [15]. In Arabidopsis, expression of the leucine-rich repeat (LRR) receptor-like kinase (RLK) *NEMATODE-INDUCED LRR-RLK1* (*NILR1*, *At1g74360*) gene is required to induce PTI responses upon NemaWater treatment obtained from *H. schachtii* and *M. incognita* [15].

The *NILR1* gene encodes NILR1 belonging to the subfamily of LRR-RLKs [15]. Like the other members of RLK, NILR1 possesses an extracellular domain (ECD), a transmembrane domain and a cytoplasmic serine/threonine kinase module [15]. The ECD of NILR1 contains 22 LRRs, which are interrupted by an island (ID) consisting of 76 amino acids [15]. Notably, Huang et al. [16] showed that the NILR1 of Arabidopsis has a high affinity for ascr#18. Based on isothermal titration calorimetry and structural analysis, the ECD of NILR1 physically interacted with ascr#18, and the ID and five C-terminal LRRs of the NILR1 were indispensable for binding [16]. However, other ascarosides, ascr#2 and ascr#3, failed to bind with the ECD of NILR1, implying that NILR1 specifically detects ascr#18 [16]. The *nilr1* mutants did not show induced defence responses upon NemaWater treatment [15], and they were more susceptible to *Pseudomonas* than wild-type Arabidopsis plants [14,16]. Therefore, NILR1 could also be involved as a co-receptor of bacterial PAMPs and other NAMPs that could trigger PTI, a hypothesis that needs further investigation. NILR1 is involved in the core branch of brassinosteroid-mediated defence signalling [17] and is widely conserved in various dicot and monocot species [14,15,16,17].

Chitin elicitor receptor 1 (CERK1) and its homologues of the LRR RLK subfamily in plants have a key role in sensing fungal cell wall-derived chitin and subsequently activating PTI [18]. In nematodes, chitin is a main component of the eggshell and pharynx [19,20]. Moreover, teeth in the nematode stylet are composed of chitin deposited during the juvenile to adult molt stages [21,22], indicating its presence during the different stages of PPN parasitism. It has been reported that plants produce a chitin-degrading enzyme, chitinase, in response to PPN attack [23,24,25,26]. Therefore, it is plausible to assume that PPN-derived chitin could be an NAMP that might be recognised by CERK1 or its homologue(s). Indeed, recent studies revealed that the expression of CERK1 was upregulated in *M. incognita*-infected cucumber (*Cucumis sativus* L.) [27] and *H. glycines*-infected soybean (*Glycine max*) [28], suggesting that CERK1 might play key roles in mediating the plant–PPN interaction. However, there is no report so far about direct binding between nematode-derived chitin and CERK1.

Reported findings have also described that Lectin RLKs (LecRLKs), members of another RLK subfamily, are involved in PPN-triggered defence responses [29,30]. To promote soybean resistance to *H. glycines* infection, two L-type LecRLKs (GmLecRK02g and GmLecRK08g) are interacting with a phosphorylated receptor-like cytoplasmic kinase (RLCK), leading to downstream defence signalling activation [29]. A G-type LecRLK, ENHANCED RESISTANCE TO NEMATODES1 (ERN1), negatively regulates PTI in Arabidopsis during RKN attack [30]. While plants deficient in the ERN1-encoding gene entail a stronger PTI and enhanced defence responses to RKN, no lesions were observed on the tissues of uninfected or RKN-infected *ern1* mutants, indicating a balanced immune response [30]. These findings suggest that adjusting negative immune regulation could enhance plant immunity without adverse effects. Despite these findings, it is unknown what these LecRLKs detect upon plant interaction with PPN.

Studies have shown that PPN detection and PTI activation in rice (*Oryza sativa*), Arabidopsis and tomato (*Solanum lycopersicum*) required BRASSINOSTEROID-INSENSITIVE 1-ASSOCIATED RECEPTOR KINASE 1 (BAK1)/SOMATIC EMBRYOGENESIS RECEPTOR KINASE 3 (SERK3) [15,31,32,33,34]. BAK1/SERK3 are common co-receptors for numerous PRRs of distinct PAMPs [32]. Interestingly, BAK1 and NILR1 interact with and phosphorylate each other in vivo [17] (Figure 1). The importance of this interaction was further evidenced by the lack of restoration of the triple mutant *bak1-8 serk1-4 bkk1-1* phenotype by NILR1 overexpression [17]. These findings showed that NILR1 requires the participation of SERK/BAK family members for its function. PTI responses induced in sweet potato (*Ipomoea batatas* Lam.) upon RKN infection involves BAK1-related signalling through respiratory burst oxidase homologues (RBOHs), calcium-dependent protein kinases (CDPKs) and MAPKs [35], implying that BAK1 activates multiple defence signalling pathways upon nematode recognition (Figure 1).

### 2.2. Intracellular Transmission of the Signal

#### 2.2.1. Receptor-Like Cytoplasmic Kinases (RLCKs)

RLCKs are important signalling proteins that connect PRR activation to downstream intracellular signalling modules, such as ROS production, calcium ion (Ca^2+^) influx, and MAPK activation [37]. A number of RLCKs, mainly members of the RLCK-II subgroup, including BOTRYTIS-INDUCED KINASE1 (BIK1) and PBS1-LIKE1 (PBL1) in Arabidopsis and RLCK118, RLCK176 and RLCK278 in rice, have been reported as key players in transducing signals from various PRRs through direct association and phosphorylation [38]. As described above, PRR co-receptor *BAK1* expression is highly induced in plants upon PPN infection. PPN-activated BAK1 interacts with and phosphorylates BIK1, which in turn phosphorylates the respiratory burst NADPH oxidase D (RBOHD), leading to an ROS burst [32,39]. Overexpression of *BIK1* leads to enhanced transcript levels of *MPK3* during the soybean defence response to *H. glycines* [40]. Zhang et al. [29] demonstrated that soybean CDG1-LIKE1 (GmCDL1), an RLCK that is a homologue of PBL7, plays a crucial role in modulating enhanced soybean defence during *H. glycines* early infection. The aforementioned LecRLKs interact with and phosphorylate GmCDL1 at Ser-234 and Thr-235 [29], while MAPK-mediated phosphorylation at Thr-372 is also necessary for its function in establishing enhanced defence responses [29].

#### 2.2.2. ROS Production

ROS generation is one of the earliest plant responses after detecting pathogens/pests, including PPNs. Elevated levels of ROS can cause oxidative damage to plant tissues by inducing protein and DNA damage, lipid peroxidation and membrane disruption [41], and therefore, plants use various cellular mechanisms to regulate the level of ROS. These include enzymatic [42,43,44,45] (Figure 1) and non-enzymatic antioxidants such as glutathione [46,47,48,49], tocopherols [50,51] and ascorbic acid [45].

The ROS burst activated upon attack has two known roles in plant defence: (1) direct toxic effects on multiple pathogenic organisms and (2) a signalling role within the plant. In resistant plants, the strong ETI response can be associated with a so-called hypersensitive response (HR), which is characterised by rapid cell death at the site of infection [8,27,52]. Prolonged accumulation of H_2_O_2_, the most stable ROS, plays an important role in activating HR-mediated defence mechanisms [53]. *M. incognita* infection, for example, triggers apoplastic H_2_O_2_ accumulation in the roots of tomato with a resistance gene, *Mi-1.2*, which eventually establishes a localised HR to arrest nematode development [8]. Moreover, the activities of H_2_O_2_-generating enzymes (superoxide dismutases, SODs) were enhanced, whereas those of H_2_O_2_-scavenging enzymes (such as ascorbate peroxidase (APX) and catalase (CAT)) were diminished during the early interactions of resistant host plants with PPNs [52,54,55].

But even a minor induction of ROS, as detected upon PTI activation for example, is instrumental in plant defence. That is because another key function of ROS in plants is that they act as signalling molecules to induce plant defence mechanisms by propagating and amplifying intercellular and intracellular defence signals. NADPH oxidases belonging to the RBOH family play crucial roles in ROS generation and signalling in plants. In response to PPN infection, RBOHB in sweet potato, tomato and Arabidopsis plays an important role in ROS production [56,57,58] and *Mi-1.2*-mediated resistance [8]. Additionally, RBOHD/F positively regulates defence responses in Arabidopsis and tomato against *M. incognita* [42,58], and RBOHD co-expresses with resistance gene *Mi-3* in tomato early upon infection by *M. incognita* [42]. Moreover, the RBOH1 (the orthologue of Arabidopsis RBOHF)-dependent MAPK pathway activation in tomato participates in the brassinosteroid-induced systemic resistance against *M. incognita* infection [59]. Interestingly, foliar application of dehydroascorbic acid (DHA), a stimulus of induced resistance (IR), causes localised H_2_O_2_ accumulation in treated rice leaves, leading to reduced *M. graminicola* infection on roots of the plant [60].

The role of ROS in plant interaction with cyst nematodes (CNs) seems more complex than in the plant root-knot nematode interaction, where ROS are involved in defence activation. More specifically, Arabidopsis RBOHD/F-mediated ROS generation activates the WALLS ARE THIN1 (WAT1) protein, which can redirect host indole metabolism, including auxin accumulation, in infected cells to promote *H. schachtii* infection [58]. In contrast, soybean RBOHG, the orthologue of Arabidopsis RBOHD, interacts with amino acid transporter (AAT_Rhg1_) and stimulates ROS production when *H. glycines* migrates through the root tissues [61]. AAT_Rhg1_ is encoded from *Rhg1-GmAAT*, which is among the genes within the soybean multicopy *Rhg1* locus that provides resistance against CNs [61,62]. Overexpression of *Rhg1-GmAAT* increases jasmonic acid (JA) levels and JA pathway genes, resulting in soybean resistance to *H. glycines* [63].

These findings highlight that the role of NADPH-mediated ROS generation during plant–PPN interaction could depend on the nematode species, host plant and its interplay with other factors such as WAT1 and AAT_Rhg1_.

#### 2.2.3. Calcium Signalling

Ca^2+^ is one of the main components of early cellular responses in mediating plant defence against pathogen infection [18]. Stimulus-induced Ca^2+^ is recognised and transduced by Ca^2+^ signalling sensors such as calmodulin (CaM) and calmodulin-like proteins (CMLs), cyclic nucleotide gated channel (CNGCs) and CDPKs [64]. These Ca^2+^ signalling mediators are involved in PTI, ETI and MAPK cascade activation and participate in SA- and JA-mediated plant defence against pathogens [64]. Accumulation of Ca^2+^ in the root cells also occurs during plant–PPN interactions [28,65,66]. It has been reported that the *CML*, *CaM*, *CNGC* and *CDPK* genes were significantly upregulated in cucumber [27], tomato [42] and sugar beet (*Beta vulgaris*) [67] at the early and late stages of nematode parasitism. Overexpression of *CML31* in rice reduces *M. graminicola*-induced gall formation by restricting the DNA binding ability of the *O. sativa* (Os) HIGH-MOBILITY-GROUP-BOX 1 (OsHMGB1) protein [68]. OsHMGB1 negatively regulates rice immunity through suppressing PR gene expression [68]. Other research has demonstrated that *CDPK4* had a higher transcript level in the resistance gene (*RMc1*(*blb*))-mediated HR of potato to *M. chitwoodi* infection [65]. Thus, Ca^2+^-mediated signalling seems to coordinate different regulatory pathways in establishing the plant defence responses to PPN infection.

#### 2.2.4. Mitogen-Activated Protein Kinase Activation

MAPK activation is involved in defence signalling by inducing the expression of multiple defence-related genes and interacting with other defence signalling components [18]. The MAPK cascade is initiated by the sequential phosphorylation and activation of three tiers of protein kinases: the upstream MAPK kinase kinases (MAPKKKs or MEKKs), the middle MAPK kinases (MKKs) and the bottom tier MAPKs (MPKs) [18]. The MAPK cascade signalling is extensively studied to be crucial for plants to defend against various fungal and bacterial pathogens. However, MAPK involvement in PPN-induced defence responses is less understood. MAPK3/6 have been implicated in regulating plant defence responses against PPN infection [59,69]. The work of Huang et al. [16] shows that MPK3/6 function downstream of the asc#18-NILR1 complex. MPK3 and MPK6 are also involved in the LecRLK-induced soybean defence response against *H. glycines* infection and wounding [29], suggesting that MPK3/6 cascade signals are activated downstream of different PRRs. MPK3/6 phosphorylate and activate CDL1 only in the presence of constitutively active MKK4, which phosphorylates and activates MPK3/6 [29]. In addition, silencing *MKK4* in soybean roots increases susceptibility to *H. glycines*, indicating that the MKK4-MPK3/6 signalling cascade positively regulates soybean defence [29]. Indeed, *H. glycines* parasitism causes MAPKs expression within syncytia undergoing a defence response [40,70]. These *MAPK* genes expressed in the syncytium include *MPK2*, *MPK3-1*, *MPK4-1*, *MPK6-2*, *MPK13-1*, *MPK16-4* and *MPK20-2* [71]. In tomato, silencing of *MPK1*, *MPK2* and *MPK3* leads to increased susceptibility to *M. incognita* [59].

Experiments have shown that the expression of *MAPKs* functions in a way that links or converges onto PTI and ETI defence branches, reducing PPN parasitism [29,34,40]. For example, increased MAPK expression regulates *PATHOGENESIS RELATED1 (PR1)* and *DOESN’T MAKE INFECTIONS3 (DMI3)* genes expression in soybean [40,71]. Overexpression of *MPK3-1* results in increased levels of *serine hydroxymethyltransferase*, *reticuline oxidase* and *xyloglucan endotransglycosylase/hydrolase* (*XTH*) transcripts [40]. Each of these genes is effective in defending soybean against *H. glycines* [40,72]. In addition, heterologous expression of *MPK3-1* in cotton (*Gossypium hirsutum*) reduces *M. incognita* root galls, egg masses and second-stage juveniles production by 80.32%, 82.37% and 88.21%, respectively [73]. Moreover, enhanced MAPK signalling pathway positively regulates flavonoid biosynthesis in the cucumber–*M. incognita* pathosystem [74].

#### 2.2.5. Phytohormones Mediate Plant Defence against Nematodes

Plant defence responses to pathogen or pest infection are usually governed by phytohormones, mainly by SA, ethylene (ET) and JA. Accordingly, these phytohormones are also involved in plant defence mechanisms to PPNs parasitism (Figure 1). SA principally modulates plant defence when plants encounter biotrophic and hemi-biotrophic pathogens [75] and can modulate plant defence responses in monocot and dicot species against root, stem and foliar nematodes [42,76,77,78,79]. Nematode parasitism is often associated with suppression of the SA pathway in susceptible cultivars, suggesting that the nematode actively interferes with this pathway through effector secretion [80]. While transgenics or mutants with reduced SA levels or SA signalling and SA inhibitor-treated plants are more sensitive to PPN attack [77,81,82,83], plants with enhanced SA levels or signalling showed reduced PPN infestation [78,84]. For example, plants treated with exogenous SA or analogues, particularly benzothiadiazole (BTH), withstand PPNs infestation [42,77,85,86]. SA triggers systemic acquired resistance (SAR) responses by regulating *non-expresser of pathogenesis-related genes1* (*NPR1*) and the transcription factor (TF), WRKY45 [42,77,78]. Beet CN infection was enhanced in *NPR1* soybean mutants [87]. NPR1 and TFs induce the expression of the SA-responsive pathogenesis-related (PR) genes such as *PR1*, *PR2* and *PR5* [78,88]. NPR1 is also involved in suppressing the JA pathway, prioritising SA over JA in Arabidopsis. The expression of SA-responsive WRKY TFs is decreased in response to *M. incognita* infection in cotton [89], again suggesting that PPNs interfere with this defence pathway to allow host infestation.

JA and its derivatives, methyl jasmonate (MeJA) and JA-isoleucine, regulate plant defence responses against threats from a wide variety of necrotrophic pathogens and herbivores [90]. Importantly, it has also been reported that the JA pathway mediates plant defence responses against biotrophic nematodes, including *M. graminicola* in rice [31,91], *M. incognita* in cotton [89] and tomato [92], *H. schachtii* in Arabidopsis [93] and *H. avenae* in wheat (*Triticum aestivum* L.) [94]. Studies showed that JA accumulation is stimulated during the early stages of RKN and CN infection [42,67,89,94]. Upon JA accumulation, JA is metabolised to JA-isoleucine, which can be detected by coronatine-insensitive 1 receptor protein (COI1). This leads to degradation of JA repressor proteins containing a jasmonate zim (JAZ) domain, which subsequently triggers several key TFs like MYC2 to activate the expression of JA-responsive genes [90]. Asadi-Sardari et al. [78] reported that the expression of JA biosynthesis and signalling genes *MYC2* and *COI1* were downregulated in a highly susceptible tomato cultivar compared with a moderately resistant cultivar upon *M. javanica* infection. These observations suggest that PPNs also actively interfere with the JA pathway. A study conducted by Guo et al. [63] demonstrated that treatment with a JA biosynthesis inhibitor reduced soybean resistance provided by *Rhg1* against *H. glycines*, implying that JA might be crucial in *Rhg1*-mediated resistance to soybean CN. Foliar spraying with MeJA results in the upregulation of Arabidopsis *Histidyl-tRNA Synthetase 1* (*HRS1*), a TF gene also responding to *H. schachtii* infection, and overexpression of *AtHRS1* modulates the expression of selected JA-related genes [93]. Moreover, exogenously applied JA and MeJA enhance plant defence responses to PPN infection by boosting the activity of antioxidant enzymes [51,95] and production of proteinase inhibitors, terpenes and oxylipins [85].

The role of ET in plant response to PPN infection seems complex and might vary depending on the receptor. A report by Hu et al. [96] showed that the *etr1-3* (*ethylene receptor1-3*) mutant did not increase sensitivity to *H. glycines*, whereas the *ein2-1* (*ethylene insensitive 2-1*) mutant attracted more nematodes to soybean. In contrast, the Arabidopsis *ein2-5* mutant showed less susceptibility to *H. schachtii* [97]. Piya et al. [97] revealed that the canonical ET signalling pathway requires CONSTITUTIVE TRIPLE RESPONSE 1 (CTR1) as an ET receptor, leading to the activation of EIN2, EIN3 and EIL1, which negatively regulates the SA pathway. On the other hand, the non-canonical ET signalling pathway needs ETR1 to crosstalk with cytokinin signalling to reduce *H. schachtii* parasitism in Arabidopsis [97]. The diverse roles of ET receptors are also detected during interactions between RKN and plants. For instance, Mantelin et al. [98] elucidated that ET receptor ETR3, but not ETR1 and ETR2, was responsible for the *Mi-1*-mediated resistance to *M. incognita* in tomato. Sikder et al. [82] found that the Arabidopsis mutant *etr1-3* attracted more *M. hapla*, whereas *ein2-1* did not affect *M. hapla* abundance. In another study, both *etr1-3* and *ein2-1* mutants positively affect *M. hapla* migration to Arabidopsis [99]. Upon exogenous induction by ethephon, the ET pathway is known to synergistically activate the JA pathway against RKNs, leading to enhanced defence against RKN [67,91]. Together, ET perception by receptors in plants influences ET crosstalk with other phytohormones, which subsequently affects the plant–PPN interaction outcome. However, more research is warranted to better understand its diverging roles.

These phytohormones are also described in modulating IR against nematodes. Singh et al. [100] found that ascorbate-oxidase-IR in rice against *M. graminicola* was dependent on both the JA and ET pathways. Likewise, SA, JA and ET pathways partially mediate the ascorbate-oxidase-IR in sugar beet against *H. schachtii* [101]. Moreover, dehydroascorbate DHA-IR against *M. graminicola* in rice involves the induction of SA marker genes *PR1a*, *PR1b* and *WRKY45* in the root tissues of rice [60]. Moreover, exogenously applied phytol triggers EIN2-dependent resistance to *M. incognita* in Arabidopsis [102], which is in accordance with the diproline-induced EIN2-dependent resistance establishment in rice against *M. graminicola* [103].

#### 2.2.6. Transcription Factors Orchestrating Plant Responses to PPN Infection

Transcription factors (TFs) can directly or indirectly play important roles in plant resistance against biotic stresses, primarily through regulating the expression of defence-related genes. The TF families, including WRKYs (WRKY domain protein), MYBs (R2R3-type MYB domain protein), bHLH (basic helix-loop-helix domain protein) and AP2/ERF (apetala 2/ethylene response factor protein), are recognised to play crucial roles in plant responses to PPN infections [104,105]. Studies have shown that TFs can act positively and negatively in transcriptional reprograming during plant responses against PPN parasitism. For example, the TF *PUCHI* gene in Arabidopsis was upregulated post-RKN infection and promotes giant cell development by regulating very long chain fatty acid biosynthesis [106]. Likewise, the *M. incognita*-induced expression of *ERF115* and *PHYTOCHROMEA TRANSDUCTION1 (PAT1)* TF genes are involved in keeping the gall functional until maturation and positively affect nematode reproduction [107]. In addition, Arabidopsis deficient in atypical TF *DP-E2F-like 1* (*DEL1*) showed an enhanced resistance to *M. incognita* infection and also led to reduced root growth, which might be due to SA accumulation in the *M. incognita*-induced galls [108].

Even within the same family of TFs, different functions have been observed. For example, overexpression of tomato *SlWRKY16* and *SlWRKY31* resulted in enhanced *M. javanica* infection [109]. On the other hand, overexpression of *SlWRKY3* in tomato led to a decrease in infection by *M. javanica*, and it was linked with the activation of lipid-, SA- and indole-3-butyric acid-mediated defence signalling [110]. Similarly, *M. incognita* inoculation led to the continuous upregulation of *SlWRKY80* in the roots of the resistant tomato cultivar carrying the *Mi-1* gene, suggesting that *SlWRKY80* plays an important role in the *Mi-1*-mediated disease resistance pathway [111]. A virus-induced gene silencing assay confirmed that *SlWRKY80* acts as a positive regulator in tomato resistance to RKNs [111].

In another study, *H. schachtii*-induced expression of TF WUSCHEL-RELATED HOMEOBOX 11 (WOX11), which functions downstream of JA-signalling via ERF109, causes increased auxin levels and secondary root formation in Arabidopsis, reducing nematode parasitism impact [112]. Similarly, Arabidopsis TF TEOSINTE BRANCHED/CYCLOIDEA/PROLIFERATING CELL FACTOR-9 modulates root growth adaptation during *H. schachtii* infection by regulating the expression of genes involved in ROS-related processes [113]. Silencing of the basic leucine zipper (bZIP) TF *TGA1a* resulted in a decreased *Mi-1*-mediated resistance to *M. javanica* in tomato [114]. Furthermore, plants impaired in TF *AtMYB12* gene expression showed hypersusceptibility to *M. incognita* infection accompanied by affected flavonoid biosynthesis in *AtMYB12* knockout lines [115]. These TF genes could help to develop engineered plant genotypes with improved performance in response to PPN infection. It is highly likely that PPNs use effectors to actively modulate the expression of these TFs genes [116], and thus, identifying TF-effector interaction partners could allow us to develop strategies abolishing these interaction and hence inhibiting nematode infection.

TF-mediated suppression of PPN infection is also triggered by IR stimuli such as β-aminobutyric acid (BABA), DHA, BTH and piperonylic acid (PA). *WRKY45*-RNAi plants showed impaired DHA-IR, implying WRKY45 functions downstream of the DHA-activated SA pathway to mediate rice resistance against *M. graminicola* [60]. In BTH-treated rice, WRKY45 promotes priming of diterpenoid phytoalexin biosynthetic genes [117]. In addition, CRISPR-Cas9 knock-out rice lines impaired in the diterpenoid phytoalexin factor, a bHLH TF [118], showed enhanced susceptibility to *M. graminicola* [119]. Higher expression of various TFs, including bHLH, MYB, ERF and zinc finger proteins was also detected in rice roots treated with BABA, BTH and PA [120]. Interestingly, these stimuli are effective in inducing the natural plant immune system, preventing PPN infection in different plant species. Therefore, they could be adopted by farmers to integrate in PPN management programs.

## 3. Epigenetics in the Plant–Nematode Interaction

Epigenetics refers to processes that lead to stable changes in gene expression across cell divisions without altering the underlying DNA sequence. Well-known epigenetic mechanisms include DNA methylation, histone modifications and regulation by non-coding RNAs (ncRNA) (see Box 1). Epigenetics plays a crucial role in determining the 3D structure of chromatin, which in turn affects DNA accessibility to the transcriptional machinery [121]. Hence, chromatin structure significantly contributes to the transcriptome of a cell under both normal and stress conditions. Therefore, the association between epigenetics and plant defence responses has become of particular interest in studying plant–pathogen interactions. In particular, PPNs, which rely on giant cells or syncytia for survival, leverage epigenetic mechanisms in their pathogenic interactions. To establish these highly specialised and differentiated nematode feeding sites, massive changes in gene expression and hence chromatin structure are required [122].

Box 1Plant epigenetics   **DNA methylation** generally comprises the addition of a methyl group to cytosine bases in the DNA and is mainly regarded to occur in CpG contexts. This is the case for the majority of cytosine methylation in animals [123] but does not reflect the situation in plants. Plant DNA can be methylated in any cytosine context, with the most studied being the CG, CHG and CHH motifs (H representing A, T or C) [124]. DNA methylation generally maintains gene transcription in a repressive state when located in gene promoters [125], while gene body methylation seems to be linked to epigenetic variation in gene expression [126,127,128]. In plants, de novo DNA methylation is mediated by the **RNA-directed DNA methylation (RdDM) pathway**. In this pathway, RNA polymerase IV transcripts are copied into long non-coding dsRNAs and subsequently processed by DICER-LIKE 3 (DCL3) into small interfering RNAs (siRNAs). These siRNAs are then loaded onto ARGONAUTE 4 (AGO4) and are guided towards nascent scaffold transcripts formed by polymerase V at the targeted region for DNA methylation. Sequence complementarity results in the recruitment of the DNA methylation machinery, establishing new patterns in all of the sequence contexts [129]. This pathway exemplifies the cooperation between different epigenetic mechanisms, in this case DNA methylation and ncRNA regulation.   The nucleosome, the basic, repeated unit of chromatin, consists of 147 base pairs of DNA wrapped around a **histone** octamer. This octamer includes two copies of four core histones: H2A, H2B, H3 and H4 [130]. Chromatin remodelling and higher order structure largely depend on the linker histone H1, which associates with the DNA between two nucleosomes [131]. Histone tails are targeted for a variety of **post-translational modifications** including acetylation, methylation, phosphorylation, ubiquitination, sumoylation, carbonylation and glycosylation [132]. These modifications can directly affect the chromatin and DNA interactions or can act by recruiting “readers”, thus regulating gene expression by altering the nucleosome positioning and stability [133]. Predicting the exact responses in terms of increased or decreased gene expression can be challenging as the response is fine-tuned by an interplay of different histone marks. Generally, acetylation is regarded to loosen the chromatin state by negatively affecting the nucleosome interactions. Conversely, the other most common mark, histone methylation, can cause either more tightly or loosely packed chromatin [132,134]. Furthermore, chromatin remodelling can also be driven by ATP-dependent chromatin remodellers. These proteins use the energy stored in ATP to modify the interaction between the DNA and histones to relocate or dissociate nucleosomes or catalyse the incorporation of histone variants [135]. Hence, these remodellers play an important role in the final fine-tuning of the chromatin structure.   **NcRNAs** are transcripts that are not translated into proteins but exert their function as an RNA. NcRNAs are typically divided into two classes: small RNAs (smRNAs) of lengths less than 40 nucleotides (nts) and long ncRNAs (lncRNAs) of lengths longer than 200 nts. Based on their biogenesis and function, smRNAs can be further subdivided into two principal classes: microRNAs (miRNAs) and small interfering RNAs (siRNAs) [136,137]. miRNAs are 21–22 nts long and are formed by processing precursor RNAs folded into a hairpin [138]. These RNAs are important in post-transcriptional gene silencing (PTGS), not only by cleaving and degrading mRNA as part of the RISC complex, but also by hampering translation [139]. siRNAs, on the other hand, are 21–24 nts long and are produced from double-stranded RNA precursors originating from hybridisation of complementary RNA strands or de novo synthesis from single-stranded RNA by RNA-dependent RNA polymerases (RDRs) [140,141]. This class of RNAs contributes significantly to transcriptional gene silencing by the RdDM pathway but also acts through PTGS. Furthermore, siRNAs often originate from transposable elements, partially explaining how transposable elements can influence gene expression [142]. LncRNAs can regulate gene expression in a variety of ways. Today, there are at least four well-known mechanisms for lncRNA regulation: histone/chromatin modification, transcriptional regulation, miRNA target mimicking and post-transcriptional alterations [143]. LncRNAs can regulate targets both in cis and trans.

### 3.1. DNA Methylation in the Plant–Nematode Interaction

Changes in DNA methylation have frequently been linked to plant defence responses. For example, infection of Arabidopsis with the bacterial pathogen *Pseudomonas syringae* pv. *tomato* (*Pst*) DC3000 leads to DNA hypomethylation in the leaves 24 h post infection, specifically in genomic regions associated with plant defence genes and at (peri)centromeric regions, while methylation-deficient mutants are less susceptible to this pathogen [144]. Hypomethylated Arabidopsis mutants (*cmt3*, *ddm1*, *drd1* and *nrpe1*) are also more resistant to the oomycete biotrophic pathogen *Hyaloperonospora arabidopsis* but not to the necrotrophic pathogen *Plectosphaerella cucumerina* [145]. Furthermore, rice treated with the chemical demethylating agent, 5-azadeoxycytidine, was found to be more resistant to *Xanthomonas oryzae* pv. *oryzae* [146].

Similar observations have been noted in plant–nematode interactions. Arabidopsis roots infected with *H. schachtii* were shown to undergo a widespread decrease in DNA methylation levels [147]. In soybean roots, *H. glycines* infection resulted in the overrepresentation of hypomethylated regions. Specifically, in the syncytium, genes mainly affected in the CG context seemed to show transcriptional effects. Considering that CG changes mainly took place in gene bodies and that gene body methylation is important in determining epigenetic variability and the transcriptional state of a gene, this indicates that the syncytia cells are epigenetically reprogrammed upon infection [126,148]. A similar hypomethylation response was observed during early gall induction by *M. graminicola* in rice. Here, loss in methylation was mainly observed in the CHH context of promotor regions at three days post infection [149]. Four days later in infection, Atighi et al. [149] linked these observations to altered gene expression, indicating that the loss in CHH methylation might prime the ET-dependent defence response. Furthermore, increased resistance to nematode infection was observed in RdDM mutants. In Arabidopsis, the *rdr2/rdr6* double and *dcl2/dcl3/dcl4* triple mutants were less susceptible to *M. javanica* [150]. In rice, the *dcl3b*, *ago4a/b* and *drm2* mutants were all less susceptible to *M. graminicola* infection [149].

Given that PPNs are master manipulators of plants, it is important to determine whether this demethylation results from plant defence mechanisms or nematode pathogenicity. Atighi et al. [149] showed that treatment of both rice and tomato with NemaWater and flg22 (a bacterial PAMP) resulted in global DNA hypomethylation, indicating that this is a nematode-independent response. Furthermore, DNA hypomethylation and downregulation of two methyltransferase genes (*CMT2* and *DRM5*) was observed in the resistant tomato cv. Rossol–*M. incognita* incompatible interaction. Contrarily, in the tomato compatible interaction, the DNA was found to be hypermethylated and the methyltransferases were upregulated [151]. Treatment of rice with azacytidine, a chemical DNA demethylating agent, resulted in resistance to *M. graminicola* infection [147], while seven-day-old giant cells showed increased expression of the RdDM pathway [152]. Together, these results indicate that DNA hypomethylation plays an important role as a plant defence mechanism, while the parasitic nematodes are capable of hijacking this system (Figure 1).

### 3.2. Histone Modifications in the Plant–Nematode Interaction

The link between histone modifications and plant defence has been frequently studied. However, the vast variety of modification types and positions, along with their interactions, makes histone modifications less understood compared with DNA methylation. Infection of *Paulownia fortunei* with phytoplasma was shown to be linked to large changes in histone H3 lysine 9 acetylation (H3K9ac) and H3K4 and H3K36 trimethylation (H3K4me3 and H3K36me3) [153]. Application of BTH induced enrichment of H3K9ac and H3K4me3 in the promotor of PR1 [154]. Bean infected with *Uromyces appendiculatus* showed association of H4K12ac and H3K9me2 with defence genes [155]. Furthermore, Singh et al. [156] found that when challenging Arabidopsis with diverse abiotic stresses, such as heat, cold and high salinity, these plants were more resistant to *Pst* DC3000 *hrcC*. They traced this effect back to the enrichment of H3K9ac, H3K14ac, H3K4me2 and H3K4me3 in several PTI-responsive genes. These marks are linked to transcriptional activation and resulted in a more open chromatin state of these defence genes. This clearly demonstrates the role of histone modifications as important mediators of plant stress responses in general.

In young *M. graminicola*-induced galls in rice, levels of H3K9ac and H3K27me3 were unilaterally increased, while the levels of H3K9me2 were unilaterally decreased. These changes were generally associated with plant defence genes [157]. Furthermore, histone modifying enzymes were found to be differentially expressed in these giant cells [152]. Next to contributing to the plant defence response, histone marks can also be manipulated by nematodes to their advantage. In the Arabidopsis–*H. schachtii* interaction, the effector 32E03 was shown to interact with histone deacetylase 1, leading to increased levels of histone H3 acetylation at rDNA regions and subsequent changes in transcriptional activity of rRNA genes in the syncytium. Interestingly, the plant manages to detect these changes as well, as overstimulation of rRNA expression resulted in silencing of these genes by RdDM [158]. In the same interaction, the cyst nematodes activated *miR778* expression to post-transcriptionally silence the H3K9 methyltransferases *SU(var)3-9 homolog 5 (SUVH5)* and *SUVH6*. These methyltransferases were shown to be important in the transcriptional reprogramming to respond to nematodes but also to develop the syncytia through deposition of the repressive H3K9me2 mark. It was shown that under infection, SUVH5/6 show a preference for targeting protein coding genes, while under uninfected conditions, they prefer to target transposable elements [159]. Hence, upon infection, this could result in the transcriptional activity of transposable elements by decreased H3K9me2. Such transposon activation has been linked to genes contributing to syncytium formation in the Arabidopsis–*H. schachtii* interaction, causing a differential expression of genes located 5 kb from differentially expressed transposons [160]. In Arabidopsis, *H. schachtii* could epigenetically target these transposons to reprogram the cell towards a syncytium. It remains the question of whether other PPNs could use similar strategies. Together, this shows that histone marks are severely impacted by the plant–nematode interaction. However, considerable effort is still needed to untangle this process.

### 3.3. ncRNAs in the Plant–Nematode Interaction

In plant–nematode interactions, lncRNAs have largely remained in the background of research. However, they possess significant regulatory potential and have been linked to defence against pathogens. For example, the lncRNAs *ELENA1* and *ALEX1* have been shown to be positive regulators of defence against *Pst* DC3000 and *X. oryzae* pv. *oryzae*, respectively [161,162]. In the soybean–*H. glycines* interaction, 384 lncRNAs were identified. These lncRNAs were predicted to be related to various nematode stress responses, either through cis or trans interactions [163]. Similarly, in the rice–*M. graminicola* and tobacco–*M. incognita* interactions, 425 and 565 lncRNAs were identified, respectively [164,165]. In the rice–*M. graminicola* interaction, the lncRNAs were linked to the regulation of signalling domain-containing proteins, indicating the broad regulatory potential of these RNAs. Furthermore, 44% of these lncRNAs showed overlap with differentially hypomethylated regions, indicating that these lncRNAs might be important in reprogramming the DNA methylation status of the surrounding genes [165].

siRNAs play an important role in gene silencing through the RdDM DNA methylation pathway and originate largely from transposable elements. Considering that transposable elements are largely demethylated and activated upon stress, this is likely to give rise to increased siRNA production [149,166]. Indeed, in the Arabidopsis–*M. incognita*, –*M. javanica* and –*H. schachtii* interactions and in the rice–*M. graminicola* interaction, 23–24 nts siRNAs were found to be the most responsive ncRNAs to nematode infection and were strongly upregulated in galls and syncytia [147,150,165,166,167,168]. Similarly, in the tomato–*Globodera rostochiensis* interaction, 24 nts siRNAs showed the largest variety in sequences amongst the ncRNAs [169]. Specifically in rice, 3739 siRNAs were responsive to *M. graminicola* infection, showing the large diversity and importance of this response [165]. These siRNAs are expected to primarily regulate genes through DNA methylation of promoters, gene bodies and transposons associated with genes [147,150,165,167]. The question remains, however, to what extent this is a nematode- or plant-induced response. PPNs could stimulate siRNAs to suppress plant defence responses while also promoting reprogramming of their feeding sites. Alternatively, as DNA hypomethylation seems to be an important plant elicited response, transposon hypomethylation and the associated siRNA production could also be a plant response mediating fine tuning of the erased DNA methylation marks to develop a directed immune response against nematodes. Future studies could aim to unravel the origins of this siRNA burst and its effects on plant defence.

In the plant–nematode interaction, miRNAs have been found to largely target transcription factor, signalling and defence genes. This was evidenced in the Arabidopsis–*M. javanica*, tomato–*G. rostochiensis*, rice–*M. graminicola* and peanut–*M. incognita* interactions and shows the broad regulatory potential of these miRNAs [165,168,169,170]. In both the Arabidopsis–*M. javanica* and rice–*M. graminicola* interaction, miRNAs were found to be mainly downregulated at early time points in the infection, namely at three days post-infection [165,168]. However, at later time points in infection such as seven and ten days, and 35 days post infection, for tomato–*G. rostochiensis* and soybean–*H. glycines* interaction, respectively, mainly upregulation of miRNAs was observed [169,171]. In the tomato syncytia, this was coupled to a decrease in miRNA variation over time, indicating that reprogramming occurs early on and that these changes are mainly sustained by the expression of a couple of miRNAs [169].

A couple of miRNAs have been studied in detail during the plant–nematode interaction. As a first example, *miR319* was shown to target the TEOSINTE BRANCHED1/CYCLOIDEA/PRO-LIFERATING CELL FACTOR 4 (TCP4) TF in the tomato–*M. incognita* interaction. Upon infection, *miR319* was upregulated resulting in decreased expression of *TCP4* in the roots. It was shown that through this targeting of TCP4, *miR319* decreases both the JA levels and biosynthesis genes expression in the root [172]. As JA is a key hormone in defence to root-knot nematodes, the *miR319*/*TCP4* module clearly suppresses the plant defence response [173,174]. This underlines the importance of miRNAs targeting TFs as pivotal regulators of the defence response. Contrarily, miRNAs can also be used by the nematode to induce their feeding sites (Figure 1). For example, *miR369* is involved in the syncytium formation-to-maintenance transition, *miR858* in transcriptional reprogramming and *miR827* in suppression of defence signalling uponCN infection [160,175,176,177]. For RKNs, *miR172* is involved in cell fate differentiation, *miR390* and *miR319* in modulation of hormone signalling and *miR408* and *miR398* in the copper deprivation response, which was proven to be essential in establishing giant cells in Arabidopsis and tomato [168,172,178,179]. A specific example of a target important in feeding site development is the auxin response factor (ARF) gene family. In both the Arabidopsis–*M. javanica* and –*H. schachtii* interactions, the miRNAs *miR390* and *miR160.15*, respectively, are upregulated and target ARF genes. This targeting was found to be important for feeding site formation as these miRNAs were expressed in early feeding sites, and inhibition of the ARF targeting resulted in resistance [168,180]. ARF targeting was also found to be important in the tomato– and cotton–*M. incognita* interactions. Here, reduced *miR167* expression upon infection resulted in increased *ARF8* expression [181,182]. In tomato, the *arf8a/b* mutants were less susceptible, indicating the importance of hijacking the auxin response of the plant [181]. This is not surprising given that auxin accumulates in nematode feeding sites and plays an important role in their formation [180,183,184]. Interestingly, in the *M. javanica* interaction, *miR390* was found to target the *TAS3* gene, resulting in secondary TAS3-siRNA, showing that nematodes are capable of inducing siRNAs in the plant [168]. A similar siRNA producing system was found in the soybean–*H. glycines* interaction [185]. Moreover, as described before, in the Arabidopsis–*H. schachtii* interaction, *miR778* was shown to target the SUVH5/6 methyltransferases, resulting in decreased preference for transposable elements under infected conditions [159]. Given that the H3K9me2 marks are depleted from transposons, they might become active, triggering the generation of siRNAs, allowing the nematode to potentially further manipulate the plant. In the Arabidopsis–*M. javanica* interaction, *miR166* was downregulated in galls [168]. *miR166* regulates shoot apical meristem and floral development in parallel to the WUSCHEL–CLAVATA pathway [186]. CNs have been shown to interfere with this pathway through secretion of CLAVATA-LIKE ELEMENTS, allowing syncytia development [187,188]. These results clearly show the importance of nematode-induced miRNAs targeting developmental processes.

### 3.4. Intergenerational Acquired Resistance in the Plant–Nematode Interaction

Epigenetics lies at the basis of memory development and inter- or transgenerational adaptation [189,190,191,192,193]. In the rice–*M. graminicola* interaction, it was shown that parent plants infected with nematodes can pass on stress memory to the next generation, leading to more resistant offspring against both *M. graminicola* and *Pratylenchus zeae* [194]. Interestingly, in uninfected offspring, this memory response manifested as a downregulation of immunity genes. However, upon infection, these plants responded strongly by inducing these memory genes. This generated a resistance phenotype by the so-called “spring-loading” of genes, triggering larger relative differences in gene expression in the remembering offspring, resulting in a stronger perceived immune response. Both the important JA and ET pathways were spring-loaded, leading to enhanced resistance. Furthermore, it was shown that the RdDM enzyme DCL3a was essential in creating this memory, indicating the importance of both DNA methylation and siRNAs in establishing this phenotype [194]. The large expression burst of siRNAs in the early nematode infection could potentially be a major contributor to the development of inter/transgenerational memory [147,150,165,167,168,169].

## 4. Plant Cell Wall Involvement in Defending against PPNs

The plant cell wall is the first physical and defensive barrier against pathogens, so all PPNs must overcome it in order to parasitise successfully. PPNs penetrate the host cells by puncturing the cell wall using their stylet and secreting modifying proteins. One of the proteins secreted by PPNs, polygalacturonase (PG), macerates the plant cell wall by hydrolysing homogalacturonan, which is a major component of pectin [6]. This promotes CN infection in plants [195]. Plants respond against PG activity by generating PG-inhibiting proteins (PGIPs), which inhibit PG-mediated homogalacturonan degradation leading to oligogalacturonides elicitor accumulation and subsequently increased resistance to CN infection [6,195]. A number of PGIPs are identified in the genomes of various important crops [196]. Arabidopsis PGIP1 mediates resistance to *H. schachtii* infection through activating plant camalexin and indole glucosinolate pathways [195]. Acharya et al. [196] found that soybean PGIP11 (GmPGIP11), not GmPGIP1, is involved in the defence response of soybean against *H. glycines* parasitism, indicating a level of specificity. Both GmPGIP1 and GmPGIP11 are predicted to have signal peptides, O- and/or N-glycosylation, and undergo secretion into the apoplast [196]. Glycosylation plays crucial defensive functions. GmPGIP11 and GmPGIP1 have distinct predicted N-glycosylation sites [196]. Overexpression of *GmBAK1-1*, *GmBIK1-6*, *GmNDR1-1* and *GmRIN4-4* led to increased relative transcript abundance of *GmPGIP11*, implying that both PTI and ETI components affect *GmPGIP11* expression [197].

Plant cell wall modification and reinforcement have also been implicated as an effective defence against PPNs [53]. Xyloglucan endotransglycosylase/hydrolase (XTHs) are plant cell wall enzymes that are involved in the modification of the xyloglucans, a component of hemicellulose present in the cell wall [198,199]. XTHs modify and restructure cell walls through the cutting and rejoining of xyloglucan chains to interconnect the microfibrils of cell walls [198,199]. XTH-mediated xyloglucan modification is considered to play a crucial role in different processes such as plant growth, development, fruit ripening and signalling [199]. *XTH* gene expression was specifically induced in soybean syncytia undergoing resistance reaction against *H. glycines* [87,200]. A transcriptomic analysis shows that infection with *M. arenaria* induces upregulation of *XTH* in Turkey berry (*Solanum torvum*) [201]. Soybean roots overexpressing *XTH43* gain the ability to suppress *H. glycines* parasitism [39]. Likewise, heterologous expression of a soybean *XTH43* in cotton impaired the parasitism of *M. incognita* [202]. The results of Niraula et al. [203] demonstrate that XTH43 activity restricts cell expansion and reinforces cell walls by remodelling xyloglucan or incorporating newly synthesised and secreted xyloglucan during the defence response of soybean against *H. glycines*. These findings indicate that XTH-mediated cell wall modification leads to the formation of a penetrative barrier against infection. Furthermore, oligosaccharides derived from the partial hydrolysis of xyloglucans act as DAMPs that could activate PTI [204,205]. Currently, the contribution of xyloglucan-derived oligosaccharides on plant–PPN interaction is not well understood. It would be intriguing to explore whether nematode infection triggers the production of xyloglucan-derived oligosaccharides and if these oligosaccharides play a role in facilitating plant basal defence responses.

Lignin and suberin form an apoplastic transport barrier, limiting the movement of water and nutrients and protecting plant cells from pathogen invasion. They are deposited as suberin lamellae and Casparian strips (CS) in the exodermal and endodermal cell walls of roots [53,206]. Investigations on incompatible interaction between plants and PPNs showed that suberin and CS confer pre-infection host plant resistance to reduce PPN penetration [207,208]. Histological and biochemical analysis demonstrated enhanced lignin and suberin deposition in the epidermal and exodermal cells of the host root system during the later stage of infection, suggesting reinforcement of structural barriers [209,210]. The endodermis controls nutrient flow into feeding sites, limiting parasitic nematodes development into adult females, thus enhancing post-infection resistance [206]. During the later stage of PPN infection, suberin biosynthesis genes were significantly upregulated in the endodermis and exodermis of the host root system [201,209,211]. The expression of these genes is probably triggered by wounding due to nematode infection [201,206]. In addition, genes involved in lignin biosynthesis were significantly expressed in the roots of resistant host plants such as Turkey berry [201], sweet potato [212], rice [105,213], soybean [104] and pine trees (*Pinus* spp.) [214] against *M. arenaria*, *M. incognita*, *M. graminicola*, *M. incognita* and *Bursaphelenchus xylophilus*, respectively. Taken together, the synthesis of lignin and suberin is induced in resistant plants [87,200,215], while PPN-induced lignification and suberisation is generally more intense in the infected plants compared with the uninfected plants and in resistant genotypes compared with susceptible genotypes [84,201,208,209].

The accumulation of lignin and suberin in the endodermis and exodermis at the feeding site may suggest that lignification and suberisation are inducible post-penetration mechanisms of plant resistance to PPN infection. Importantly, IR stimuli such as BABA, sclareol, chitosan and thiamine effectively trigger lignin accumulation and suppress PPN infection [53,66]. In addition, tomato treatment with BTH enhances the expression of lignin synthesis-related genes and lignin accumulation at the feeding site following *M. incognita* infection [210]. Furthermore, Desmedt et al. [120] observed that rice leaves treated with IR stimuli (BABA, BTH, DHA and PA) showed significant induction of lignin and suberin synthesis-related genes, which is correlated with reduced development and reproduction of nematodes [210]. In conclusion, accumulation of lignin and suberin reinforces the cell wall, which most likely impedes both nematode penetration as well as the flow of nutrients to nematode feeding sites, reducing plant susceptibility to attack.

## 5. Metabolic Changes and Anti-Nematode Compounds Production

Plants release various active phytochemicals into the rhizosphere through their root system, which can act against nematode attacks and play a role in plant defence. A better understanding of how these metabolites interact with nematodes can help us develop new strategies to manage nematode infestations. In response to nematode invasion, diverse secondary metabolites such as phenolic acids, terpenoids, organosulphur compounds, benzoxazinoids, alkaloids, saponins and glucosinolates are also produced inside the root system as a chemical defence mechanism, reviewed in [4,53]. Some important secondary metabolites involved in plant–PPN interaction are discussed under the following sections.

### 5.1. Glycine Betaine

Glycine betaine (GB), Figure 2A, is a quaternary ammonium compound that some plants produce in their chloroplasts using choline or glycine as initial metabolites [216]. GB functions as an osmoregulator, stabilises enzymes and protein complexes and helps maintain the integrity of membranes, protecting them from stress-induced damage [216,217]. GB likely activates CDPKs and MAPKs, which could boost the natural defence system by enhancing enzymatic antioxidant activity, alleviating the negative impact of uncontrolled ROS causing oxidative damage [216]. GB also appears to be involved in plant response to PPN parasitism. An increase in the level of GB was reflected in *M. incognita*-infected *Lycopersicon esculentum* seedlings [218,219]. Similarly, *M. incognita* infection and the application of JA and spermine to tomato showed enhancement in GB content as compared with control and nematode-inoculated plants [51]. Recently, Zhang et al. [220] identified an effector (MiATRR) in *M. incognita* that is significantly upregulated at the early parasitic life stage. Host-derived *miatrr*-RNAi in Arabidopsis significantly reduces the number of galls and egg masses of *M. incognita* as well as retards development and decreases the body size of the nematode [220]. Zhang et al. [220] suggested that MiATRR acts as a glycine betaine reductase, converting GB to choline, thereby promoting *M. incognita* invasion. The positive role of GB in plant defence to PPN is further confirmed by reduced numbers of galls and egg masses after exogenous application with GB [220]. But choline application enhances the numbers of galls and egg masses [220]. With respect to these findings, intensive research is needed to reveal how and to what extent GB could be involved during plant–PPN interactions. For instance, information is needed about the interplay of GB with CDPKs, MAPKs and phytohormones during plant–PPN interaction.

### 5.2. Organosulphur Compounds

Glutathione (Figure 2B) is one of the organosulphur compounds mainly synthesised by γ-glutamylcysteine synthetase (GSH1) and glutathione synthetase (GSH2) in the cytosol and plastids of plants. It mostly exists in the reduced form, GSH, possessing a free thiol group that confers its wide biological activities as an anti-oxidant, as well as in cellular signalling and detoxification of xenobiotics and heavy metals [221]. Glutathione has been frequently shown to have a crucial role in plant responses to (a)biotic stresses [222,223]. Glutathione metabolism perturbation is among the metabolic alterations caused by nematode infection in plants. For example, in a resistant wheat genotype, *P. thornei* infection led to upregulation of the glutathione pathway [224]. Similarly, a transcriptional analysis showed that glutathione metabolism was upregulated in a compatible sweet potato variety in response to *M. incognita* infection [225]. In another study, GSH activity was significantly enhanced in tomato inoculated with *M. enterolobii* [49]. Glutathione is best known for its role to regulate ROS and protein modification in stressed plants [223].

Arabidopsis loss-of-function *gsh* mutants are impaired in camalexin production during *H. schachtii* infection [226]. Camalexin accumulation was reduced in *cadmium-sensitive2* (*cad2*) and *zinc tolerance induced by iron 1* (*zir1*) mutants [224], which are known for their roles in maintaining the glutathione level [227,228]. It has been described that camalexin biosynthesis has importance in plant defence response to nematode infection [4]. GSH mitigates biotic stresses via activation of NPR1-dependent SA-mediated defence responses [229]. Thus, GSH could alleviate nematode-induced stress in plants via the accumulation of anti-nematode metabolites and PR proteins generation in addition to its role in removing ROS.

Organosulphur compounds extracted from non-host plants have also been tested for their potential in crop protection against PPNs. For example, α-terthienyl is abundant in the roots of *Asteraceae* family species (mainly *Tagetes* sp.), and it has been described as a potent nematicidal compound [5]. So far, α-terthienyl was shown to be suppressive to *P. penetrans*, *M. incognita* and *Nacobbus aberrans* [4,230]. The inhibition mechanisms of α-terthienyl are well described in a previous review from our group [4].

### 5.3. Terpenoids

Terpenoids are among the most diverse class of plant secondary metabolites identified in several plant species. Terpenes are formed through the condensation of activated isoprene units. Depending on the number of units, they can be monoterpene, sesquiterpene or diterpene [4]. Many terpenoids are involved in defence against pathogenic bacteria, fungi and insects. Biosynthesis of terpenoid phytoalexins was strongly involved in sweet potato [79], wheat [224] and rice [105] response to *D. destructor*, *P. thornei* and *M. graminicola* infection, respectively. Elsharkawy et al. [231] conducted a study on the resistance induction and nematicidal activity of four monoterpenes (carvone, cineole, cuminaldehyde and linalool) against *M. incognita* in tomato under laboratory, greenhouse and field conditions. Among these monoterpenes, carvone followed by cuminaldehyde resulted in a reduced number of egg mass, J2 and galls compared with the other monoterpenes and an infected control. In in vitro and pot experiments, a monoterpene, α-Terpinene, displayed the highest toxicity to J2 of *M. javanica* [232]. High efficacy of carvone (Figure 3a), cuminaldehyde (Figure 3b) and α-Terpinene (Figure 3c) against RKN is associated with the presence of hydroxyl or carbonyl group in these terpenoids [231], which indicates that the functional group is key in their nematicidal activity. Besides this, the spatial arrangement of atoms in the molecules may influence their bioavailability and bioactivity against PPNs [233]. In another study, indirect contact bioassays indicated that the oxygen-containing monoterpenes were more effective in causing mortality in *P. penetrans* than hydrocarbons [234]. This suggests that the presence of oxygen in monoterpenes is essential for their nematicidal activity. Furthermore, toxicity of these monoterpenes extends to various phytopathogens such as *Rhizoctonia solani*, *Asperigallus niger*, *Fusarium oxysporum*, *Sitophilus oryzae* and *Tribolium castaneum* under in vitro, greenhouse and field conditions [235,236]. This clearly demonstrates that monoterpenes could be a promising candidate for developing eco-friendly strategies in managing diverse pathogens and pests in plants.

Tomato plants treated with cuminaldehyde, carvone, linalool and cineole led to higher transcription levels of *PR1* and *PAL* genes [231]. Moreover, treatment of common bean plants with carvone, cuminaldehyde and linalool resulted in an elevated transcription level of a defence gene, *β-1,3-glucanase*, along with a significant increase in the activity of POX, PPO and CAT in the leaves [235]. These reports indicate that monoterpenes could also be used as IR stimuli. However, it remains unclear whether the monoterpenes-IR reduces plant infestations by nematodes or other pathogens. It will also be important to evaluate the durability of IR conferred by monoterpenes.

Diterpenoid phytoalexins play a significant role in both the basal and inducible defence responses of rice to nematodes. A mutant rice line resistant to *M. graminicola* shows early expression of diterpenoid biosynthesis genes [237]. Additionally, rice lines genetically impaired in diterpenoid biosynthesis showed higher susceptibility to *M. graminicola*, confirmed by a higher number of nematodes per root system [119]. Exogenous application of JA, DHA and PA triggers the biosynthesis of diterpenoid phytoalexins [91], and these IR stimuli had previously shown efficacy against *M. graminicola* [100,238]. Desmedt et al. [119] observed that rice diterpenoids released to the rhizosphere affect rice-associated nematode communities, including effects on nonphytoparasitic nematodes. This is confirmed by increased abundance of PPNs like *Pratylenchus* and *Meloidogyne* but depletion of predatory nematodes of the genus *Mononchus* in the roots of rice impaired in diterpenoid biosynthesis [119]. Furthermore, rice diterpenoids accumulate significantly in response to UV stress, heavy metal exposure and infections by pathogens like *Magnaporthe oryzae* and *X. oryzae* [239]. Together, these reports clearly prove that diterpenoids play an important role in the overall stress responses of rice.

### 5.4. Benzaldehyde

Benzaldehyde (Figure 4a) is derived from transcinnamic acid of the shikimate pathway, consisting of a single benzene ring bearing an aldehyde group [240]. Benzaldehyde has been found in various plant species [241], and its nematicidal activity was demonstrated in studies with *M. incognita* under in vitro and in vivo experiments [241,242]. It was hypothesised that benzaldehyde acts against nematodes by limiting the activity of its V-ATPase enzyme, which is involved in nematode physiological processes such as cuticle synthesis [243]. Barbosa et al. [234] found that benzaldehyde achieved full mortality on *P. penetrans* by damaging its internal tissues rather than its cuticle shape. These results indicate that benzaldehyde seems to have a different mode-of-action against RKN and migratory nematodes. The reported environmental and (eco)toxicological parameters for benzaldehyde suggest lower toxicity and higher safety of use [234].

The hydroxyl derivatives of benzaldehyde, including 2-hydroxybenzaldehyde (salicylaldehyde) (Figure 4b), 3-hydroxybenzaldehyde (Figure 4c) and 4-hydroxybenzaldehyde (Figure 4d), showed strong nematicidal activity to *M. incognita* [243,244]. Salicylaldehyde was the most active benzaldehyde followed by 3-hydroxybenzaldehyde and 4-hydroxybenzaldehyde, indicating that position 2 of the hydroxyl group in the benzene ring appears to be critical for their nematicidal activity to *M. incognita* [244]. A synergistic activity was observed when salicylaldehyde was added to 3-hydroxybenzaldehyde and to 4-hydroxybenzaldehyde [244]. It was suggested that, similar to benzaldehyde, salicylaldehyde is a redox-active compound capable of generating ROS, which may play a role in impairing the functionality of V-ATPase and consequently affecting the osmoregulation of nematodes [243].

In plants, benzaldehyde and hydroxybenzaldehyde can be further oxidised into benzoic acid (BA) and hydroxybenzoic acid (HBA), respectively, which are functionally important compounds [240,245]. For instance, the well-known plant hormone salicylic acid (2-HBA) involved in defence signalling is a derivative of salicylaldehyde through the BA biosynthesis pathway. Previously, Nguye et al. [246] purified a BA, 3,4-dihydroxybenzoic acid (3,4-DHBA, Figure 4f), from *Terminalia nigrovenulosa* bark and tested its in vitro nematicidal activity against *M. incognita*. The study showed that 3,4-DHBA treatment led to hatch inhibition and J2 mortality in a dose-dependent manner. Likewise, 3,5-dihydroxybenzoic acid (3,5-DHBA, Figure 4g) extracted from *Rubus niveus* exhibited moderate in vitro nematicidal activity against *M. incognita* [247]. Recently, Yates et al. [248] utilised a phytochemical-based seed coating method on soybean seeds, applying 2,3-dihydroxybenzoic acid (2,3DHBA, Figure 4e) and 4-hydroxybenzaldehyde to inhibit soybean cyst nematodes (CSN). 2,3DHBA significantly reduced the abundance of SCN in infected plants, and it showed no phytotoxicity. Although 4-hydroxybenzaldehyde has no phytotoxicity, its SCN reduction capability was not significantly different from the control treatments. The reduced efficacy of 4-hydroxybenzaldehyde could be related with its instability in soil.

The accumulation of shikimate-derived BA was increased in the *SlWRKY3*-overexpressing tomato infected with *M. javanica* [110]. Tomato inoculation with *M. incognita* resulted in a significant increase in 4-hydroxybenzaldehyde compared with the uninfected control [208]. Interestingly, the application of DHA, BABA and PA led to enriched benzoic acid derivatives in the rice root exudates [120], most likely serving as deterrents to PPN. Likewise, BTH-treated susceptible infected roots of tomato showed increased 4-hydroxybenzaldehyde that could play a critical role in defence to *M. incognita* [210].

Together, shikimate-derived benzaldehydes and derivatives thereof are part of the inducible biochemical defence of plants against PPNs. This makes them a promising option for developing biopesticides and implementing a more sustainable pest management strategy. However, further studies are necessary to investigate the stability of benzaldehydes and their derivatives when introduced into the environment. It is important to also evaluate how these compounds interact with other nematicides to effectively control nematodes.

### 5.5. Benzoxazinoid Compounds

Benzoxazinoids (BXs), comprising the classes of benzoxazinones and benzoxazolinones, are a set of specialised metabolites produced by plants in the family *Poaceae*, such as maize and wheat, as well as some dicots. BXs have been shown to act as PPN attractant and are also positively correlated with resistance to PPN [4]. Recently, Sikder et al. [249] found that maize plants that produced BXs selectively enhance or reduce the abundance of specific nematodes in and around their roots. This is evidenced by enriched *P. neglectus* but reduced *P. crenatus* abundance in the roots of *bx1* maize mutants impaired in BXs biosynthesis [249]. Exuded BXs are considered to have an allelopathic effect [250] and can be taken up by non-BX-producing plants and translocated to their shoots [251]. The uptake of BXs alters the composition of intrinsic secondary metabolites, in particular, flavonoids and abscisic acid in clover (*Trifolium repens* L.) [251], enhancing its resistance to *M. incognita* invasion [252]. The exact mechanisms of action of BXs in resistance against PPNs need further investigation. In addition, BXs produced by plants in defence against PPNs can accumulate initially in the PNN-infested soil, leading to the PPNs gaining tolerance to BXs toxicity. Furthermore, investigating the structure–bioactivity interaction of BXs could provide insights into their role in the plant-nematode interaction.

## 6. Conclusions

Plant parasitic nematodes (PPNs) are responsible for affecting almost all types of plants, leading to substantial economic losses due to decreased yield and quality. In response to PPN detection and invasion, plants initiate a complex defence mechanism. This involves networked signal transduction events such as reactive oxygen species burst, calcium ion influx, mitogen-activated protein kinases activation, phytohormone synthesis and transcription factor activation. These early signalling events lead to the induction of defence mechanisms including hypersensitive response, cell wall reinforcement and the production of different defensive secondary metabolites. Research has recently revealed that epigenetic modulation also plays a major role in the plant–nematode interaction. It is important in triggering the appropriate plant response to nematode infection but also forms an important tool for the nematode to evade plant defence and successfully establish feeding sites. Future research will be necessary to distinguish between plant immune responses and nematode manipulation of the plant.

Interestingly, natural plant defence systems can be induced by applying plant-originated stimuli such as dehydroascorbic acid, piperonylic acid, β-aminobutyric acid and sSA(-analogues), which can prevent PPN infections in plants. Plant secondary metabolites like glycine betaine, glutathione, terpenoids, benzoic acid and benzoxazinoids can either modulate defence crosstalk or act as nematicidal compounds. This shows that farmers can potentially use plant-derived compounds as part of PPN management programs. Given the complex nature of plant–PPN interactions, gaining a deeper understanding of plant immunity and resistance to PPNs will greatly help in the development of innovative and sustainable strategies for managing PPNs.

## Figures and Tables

**Figure 1 plants-13-02813-f001:**
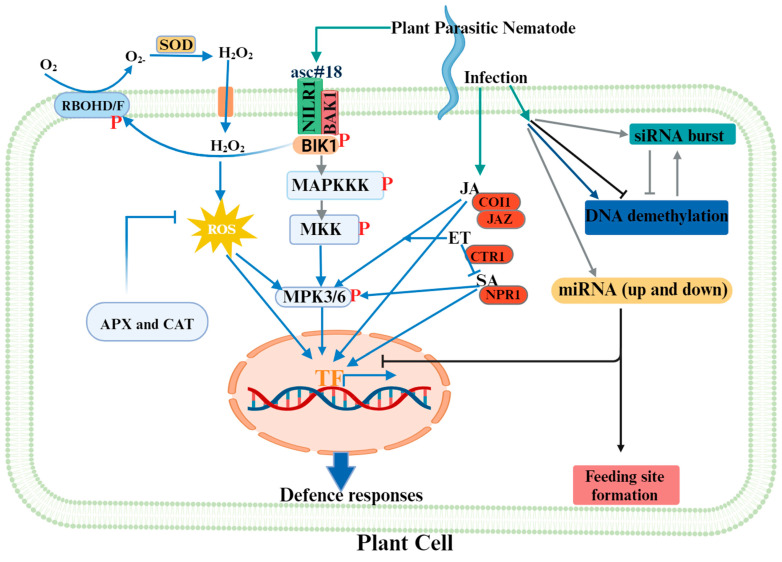
Plant-parasitic nematodes release pheromones that act as nematode-associated molecular patterns that can be detected by plants, such as Arabidopsis, activating defence mechanisms. The receptor-like kinase NEMATODE-INDUCED LRR-RLK1 (NILR1), located in the plasma membrane of Arabidopsis, detects the nematode-released pheromone ascaroside #18 (asc#18). Consequently, the kinase-active cytoplasmic region of NILR1 interacts with its co-receptor BAK1 and phosphorylates each other. When BAK1 is activated, it interacts with and phosphorylates BIK1, which then phosphorylates the plasma membrane-localised RBOHD/F enzymes, leading to a burst of reactive oxygen species (ROS) in the cytoplasm and apoplast. RBOHD/F-aided ROS generation, causing defence activation, also occurs when root-knot nematodes migrate in the root system. The role of ROS becomes more intricate when root tissues are damaged due to cyst nematode infection. More details can be found in a recent review [36]. Upon asc#18 detection, mitogen-activated protein kinase 3 (MPK3/6) functions downstream of BIK1, but MAPK cascades that link BIK1 and MPK3/6 are less understood. Plant epigenetics events play a major role in the plant–nematode interaction. DNA methylation is decreased by the plant but increased by the nematode. Small interference RNA (siRNA) expression is heavily increased in the plant upon nematode infection. siRNAs could lead to DNA methylation or could be the result of the demethylation of transposable elements. The expression of microRNAs (miRNAs) is likely misused by the nematode to inhibit transcription factors (TFs) involved in defence activation and promote the formation of nematode feeding sites. The Arrow is a positive relation, and the perpendicular line is a negative relation. Blue line: expected plant response; black line: expected nematode response; grey line: unrevealed response. APX, ascorbate peroxidase; BAK1, BRASSINOSTEROID-INSENSITIVE 1-ASSOCIATED RECEPTOR KINASE 1; BIK1, BOTRYTIS-INDUCED KINASE1; CAT, catalase; P, phosphorylation. This illustration was created with www.BioRender.com accessed on August 2024.

**Figure 2 plants-13-02813-f002:**
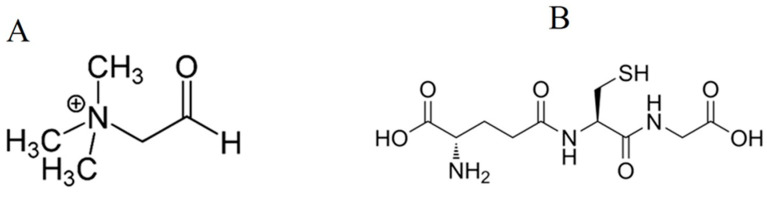
Chemical structure of metabolites. Both glycine betaine (**A**) and glutathione (**B**) are non-enzymatic antioxidants that could be produced in plants in response to nematode infection.

**Figure 3 plants-13-02813-f003:**
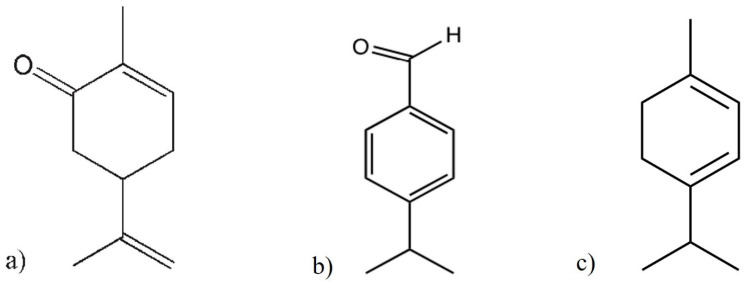
Chemical structure of monoterpenoids: carvone (**a**), cuminaldehyde (**b**) and α-Terpinene (**c**), which have strong nematicidal activity.

**Figure 4 plants-13-02813-f004:**
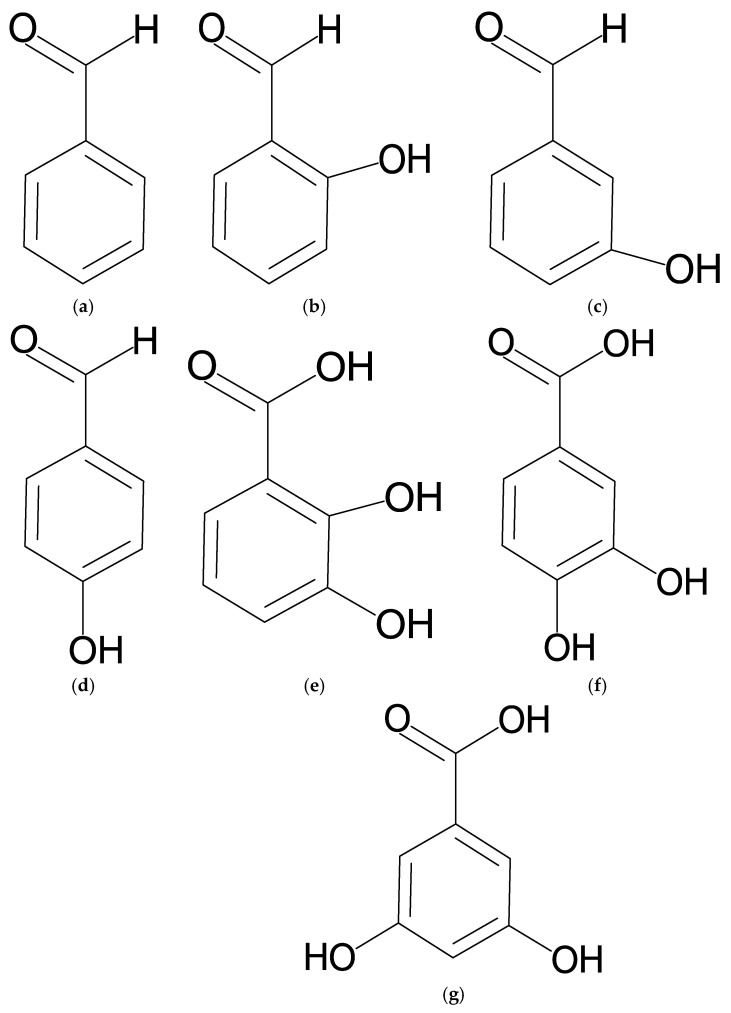
Chemical structure of benzaldehyde (**a**), salicylaldehyde (**b**), 3-hydroxybenzaldehyde (**c**), 4-hydroxybenzaldehyde (**d**), 2,3-dihydroxybenzoic acid (**e**), 3,4-dihydroxybenzoic acid (**f**) and 3,5-dihydroxybenzoic acid (**g**), which showed toxicity to plant parasitic nematodes.

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
