# Peer review of "Biochemical Defence of Plants against Parasitic Nematodes"

_plants, 2024, doi:10.3390/plants13192813_

Round 1

Reviewer 1 Report

Comments and Suggestions for Authors

Dear authors,

Your manuscript is quite good in terms of the topic, which is obviously of interest and future.

However, some suggestions for improvement are necessary to increase the value. Please find the comments on them below:

Line 7-22: The summary approach is very vague about nematode species. Please detail which group of nematodes you are referring to? Or at least to which family or category of them.

Line 57-58: Related to: <Some varieties carry nematode resistance genes, such as the Mi1.2 gene that makes 57 specific tomato cultivars resistant to root-knot nematodes> it is necessary to name exactly the species or species of nematodes referred to in this reference. I only found the summary of the reference that states this, but if you have cited the entire article, you probably also have the species involved in their study, so that the citation highlights the reality.

Line 150-151: Related to:legend from Figure 1: a)<Plant-parasitic nematodes released pheromones act as nematode-associated molecular pat-150 terns that can be detected by plants, activating defence mechanisms >mention which plant species you are referring to

Line 151-152: Related to:legend from Figure 1: <The plasma mem-brane-local-151 ized receptor-like kinase NEMATODE-INDUCED LRR-RLK1 (NILR1) detects nematode released 152 ascaroside #18 (asc#18) pheromone. >The phrase is illogical and incomprehensible. The legend was probably generated automatically, so please review and summarize the description of the legend and the connection to the scheme in a clearer way.

As a general impression, the manuscript is well documented at the level of biochemical processes, but there is no coherent flow between ideas. Try to organize the ideas by categories of processes that have continuity and a connection between them.

 Please also mention somewhere, at the end of the introduction, which are the species of plants and those of nematodes that you refer to in this review to facilitate the understanding of the one who reads the manuscript.

Kind regards,

R

Author Response

Dear Reviewer,

Thank you very much for taking the time to review this manuscript. Please find the detailed responses below and the corresponding revisions in track changes in the re-submitted files.

1. Point-by-point response to Comments and Suggestions

Comments 1: Line 7-22: The summary approach is very vague about nematode species. Please detail which group of nematodes you are referring to? Or at least to which family or category of them.

Response 1: Thank you for pointing this out. Comment taken. Therefore, we have mentioned the nematodes family that are frequently studied in this manuscript. These changes can be tracked on the first page, Abstract section, and line number 7.

Comments 2: Line 57-58: Related to: <Some varieties carry nematode resistance genes, such as the Mi1.2 gene that makes 57 specific tomato cultivars resistant to root-knot nematodes> it is necessary to name exactly the species or species of nematodes referred to in this reference. I only found the summary of the reference that states this, but if you have cited the entire article, you probably also have the species involved in their study, so that the citation highlights the reality.

Response 2: Mi1.2 is best-studied for its resistance against Meloidogyne incognita. However, more and more reports show that it provides resistance to other RKN as well. This change can be found at page number 2, line number 7-8.

Comments 3: Line 150-151: Related to: legend from Figure 1: a) <Plant-parasitic nematodes released pheromones act as nematode-associated molecular patterns that can be detected by plants, activating defence mechanisms >mention which plant species you are referring to.

Response 3: Thank you for pointing this out. We agree with the comment. Thus, the name of the plant species is now included. This change can be tracked on page number 4, line number, 157

Comments 4: Line 151-152: Related to: legend from Figure 1: <The plasma mem-brane-local-151 ized receptor-like kinase NEMATODE-INDUCED LRR-RLK1 (NILR1) detects nematode released 152 ascaroside #18 (asc#18) pheromone. >The phrase is illogical and incomprehensible. The legend was probably generated automatically, so please review and summarize the description of the legend and the connection to the scheme in a clearer way.

Response 4: Comment taken. The sentence < The plasma mem-brane-local-151 ized receptor-like kinase NEMATODE-INDUCED LRR-RLK1 (NILR1) detects nematode released 152 ascaroside #18 (asc#18) pheromone. > is revised into < The receptor-like kinase NEMATODE-INDUCED LRR-RLK1 (NILR1), located in the plasma membrane of Arabidopsis, detects the nematode-released pheromone ascaroside #18 (asc#18).> to make it more clear. This change can be found on page number 4, line number 158-159.

Comments 5: As a general impression, the manuscript is well documented at the level of biochemical processes, but there is no coherent flow between ideas. Try to organize the ideas by categories of processes that have continuity and a connection between them.

Response 5: In our manuscript we describe the biochemical events that happen upon nematode infection, in a chronological way . We start from the early detection of nematode presence in the plant (PAMPs/DAMPs at the plasma membrane) over the intracellular signaling (ROS, Ca, MPKs, hormones) to the downstream effects on epigenome, transcriptome and finally the metabolome (cell wall reinforcement, secondary metabolite production). This explanation has now been added to the text, page 2, lines 81-83.

Comments 6: Please also mention somewhere, at the end of the introduction, which are the species of plants and those of nematodes that you refer to in this review to facilitate the understanding of the one who reads the manuscript.

Response 6: We agree with the comment. Therefore, categories of plants and nematodes families which are repeatedly studied in the reviewed literature are included on page number 2, line number 78-80.

Reviewer 2 Report

Comments and Suggestions for Authors

Meresa et al. have written a truly encyclopedic review encompassing much of what is currently known about plant-based defenses against parasitic nematodes. 

I noticed a few places where some minor changes might improve the story.  For example

-Would it be worthwhile to insert a place for ethylene in their figure given that some papers do show ethylene appears to be important (line 369)?

-Pg 9, line 416: the definition of epigenetics seems too general.  It could apply to any inducible or repressible system (e.g., heat shock induced genes, beta-galactosidase induction in E. coli, etc.).  I know it is hard to define epigenetics.  Perhaps if they said “…that cause (mitotically or meiotically) heritable changes…”, they might come closer to a satisfactory definition.

-Pg 10, line 435:  This statement caught me my surprise.  The papers I have seen indicated methylation of transcribed regions had comparatively little if any effect on transcription.  The reference they cite is a review.  I think it would be nice if they instead cited some primary literature when they want to make this point.  

Continuing on this point, I think that “transcribed region” offers a more precise identification of the methylase targets (as in line 499) than “gene body”.

-Pg 12, line 555, and pg 14, line 636 refers to the effects of miRNA 778 on the regulation of methylation.  I am concerned by the suggestion that “nematodes could epigenetically target these transposons to reprogram the cell towards a syncytium”.  I didn’t see a specific statement saying nematodes induce miR778, nor did I see which nematode does this, nor did I see that this nematode induces functionally identical microRNAs in plants other than Arabidopsis.  In other words, is this a coincidence of p-hacking the right nematode and the right host, or is the nematode in question able to induce microRNAs that inhibit methylases in many/most of its hosts? 

-One thing reviewers can do is not simply report what has been written, but critique it as the original authors perhaps could not.  Meresa et al. do this well on pg 13 (line 587 and onward).  I just wish they had done a little more elsewhere.  For example, I think it might be useful to distinguish between general stress responses (as in line 533) and nematode or pathogen-specific responses as in their discussion of toxic metabolites.

-In general, this paper is well written.  There were two problems with the language that I think detracted from the reader’s enjoyment of it.  One was that definite (and occasionally indefinite) articles were absent from many places where they would likely have been inserted by native English speakers.  For example, “TheNILR1 gene” in line 97, “The ECD of”, line 102.  This probably will not cause problems for readers who have grown up with languages that lack these grammatical forms. 

-Another small class of errors was that many words were singular where the subject of the sentence was (or should have been) plural.  For example, pg 13, line 623: I think the clause should end with “M. javanica and H. schachtii interactions”.  Similarly, in line 668, it should read “plants respond”, in line 674 and line 845, “acts against nematodes”.

-Rarely, the authors used phrases that sounded a bit awkward.  For example, one usually doesn’t refer to positions on a molecule as “highly significant” as written in line 858.   “Critical” or “Very important” would convey the same message. Line 879, “could, probably” sounds redundant.  “could” would suffice.  Another example would be line 336 where the authors talk about a mutation not promoting “abundance”.  I have not heard that word used in nematode literature and think it would be better to stick to conventional terms such as “did not increase sensitivity” or phrases like that.

Comments on the Quality of English Language

 Overall, the authors might consider having a native English speaker edit this, or they can just let it go as the information is what matters.  If they do employ an editor, that person  can also remove some unwanted spaces such as between up and on in line 49 and unwanted commas such as in line 548.

In all other ways, this is likely to be a very welcomed review for those in the field.

Author Response

Dear Reviewer,

We would like to appreciate for taking the time to review this manuscript. Please find the detailed responses below and the corresponding revisions in track changes in the re-submitted files.

1. Point-by-point response to Comments and Suggestions

Comment 1: Would it be worthwhile to insert a place for ethylene in their figure given that some papers do show ethylene appears to be important (line 369)?

Response 1: Thank you for pointing this out. Yes, the ethylene pathway has an important but complex role during plant-nematode interactions, as discussed on page number 8, line number 349-354. This is inserted in the figure as well.

Comment 2:  Pg 9, line 416: the definition of epigenetics seems too general.  It could apply to any inducible or repressible system (e.g., heat shock induced genes, beta-galactosidase induction in E. coli, etc.).  I know it is hard to define epigenetics.  Perhaps if they said “…that cause (mitotically or meiotically) heritable changes…”, they might come closer to a satisfactory definition.

Response 2: You are right, we have included the stable nature of the processes across mitotic and sometimes even over meiotic cell division. Changes can be found in the submitted version on page 9 paragraph 4.

Comment 3: Pg 10, line 435:  This statement caught me my surprise.  The papers I have seen indicated methylation of transcribed regions had comparatively little if any effect on transcription.  The reference they cite is a review.  I think it would be nice if they instead cited some primary literature when they want to make this point. 

Continuing on this point, I think that “transcribed region” offers a more precise identification of the methylase targets (as in line 499) than “gene body”.

Response 3:  The review was cited because the current hypothesis includes results from various primary articles. To clarify, we added two primary articles. The hypothesis is as follows: Gene body methylation occurs frequently in constitutively expressed genes. Based on studies profiling the methylation status of genes across natural accessions, it was observed that gene body methylation could be linked to epialleles and epigenetic variation. Mechanistically, Robert J. Schmitz proposes that these changes arise from a complex interplay between CMT3-H3K9me2-IBM1. Changes in activity of either causes changes in gene body methylation which in turn results in these epialleles. Daniel Zilberman, has modelled gene body methylation across different timescales, showing that stochastic changes in gene body methylation occur over various timescales, which allows epialleles to be formed (and potentially be selected and maintained).

We chose to describe the region as “gene body” as this is the convention in epigenetic studies focusing on DNA methylation, “transcribed region” is seldomly used to describe these positions in this context.

Comment 4: Pg 12, line 555, and pg 14, line 636 refers to the effects of miRNA 778 on the regulation of methylation.  I am concerned by the suggestion that “nematodes could epigenetically target these transposons to reprogram the cell towards a syncytium”.  I didn’t see a specific statement saying nematodes induce miR778, nor did I see which nematode does this, nor did I see that this nematode induces functionally identical microRNAs in plants other than Arabidopsis.  In other words, is this a coincidence of p-hacking the right nematode and the right host, or is the nematode in question able to induce microRNAs that inhibit methylases in many/most of its hosts?

Response 4:  Changes were made to the text on page 12 paragraph 2 to include more clarity on the nematode system that was used in the references, the observed response and the fact that this has not yet been observed in other pathosystems.

Some clarification for the reviewer: The discussion of the topic kicks-off with “In the same interaction” hinting at the previous Arabidopsis-H. schachtii interaction (to avoid excessive repeating of this construction). The paper of Bennet et al. (2022) does indeed show that the nematode induces expression of miR778 in the syncytia, even more so, overexpression lines and target mimics affect the susceptibility in a coherent manner. We tried to highlight this by explicitly mentioning that the nematode induces the miR778 expression. Since other studies on this topic in other PPN-plant interactions are lacking, we have now clarified this in the text.

Comment 5:  One thing reviewers can do is not simply report what has been written, but critique it as the original authors perhaps could not.  Meresa et al. do this well on pg 13 (line 587 and onward).  I just wish they had done a little more elsewhere.  For example, I think it might be useful to distinguish between general stress responses (as in line 533) and nematode or pathogen-specific responses as in their discussion of toxic metabolites.

Response 5:  Thank you very much for pointing this out. Changes were made to the text on page 18 to resolve these concerns.

2. Response to Comments on the Quality of English Language

Point 1: Minor editing of English language required.

Response 1: The comments on the quality of English language are addressed throughout the submitted manuscript in the track change format.

Round 2

Reviewer 1 Report

Comments and Suggestions for Authors

No comment. The suggestions have been resolved.